# Health-related quality of life and psychosocial impacts of a diagnosis of non-specific genital infection in symptomatic heterosexual men attending UK sexual health clinics: a feasibility study

Rachel Hill-Tout,[1,2] Emma M Harding-Esch,[2,3] Agata Pacho,[3] Martina Furegato,[2,3] Sebastian S Fuller,[3] Syed Tariq Sadiq[1,2,3]

SSF and STS contributed equally.

[1]St Georges University Hospitals NHS Foundation Trust, London, UK
[2]Public Health England, London, UK
[3]Applied Diagnostic Research and Evaluation Unit, Institute for Infection & Immunity St George's, University of London, London, UK

**Correspondence to**
Dr Sebastian S Fuller; sfuller@sgul.ac.uk

## ABSTRACT

**Introduction** Non-specific genital infection (NSGI; non-*Chlamydia trachomatis*, non-*Neisseria gonorrhoeae*-associated urethritis) is a common diagnosis in symptomatic heterosexual men attending UK sexual health clinics (SHCs). but little is known about the psychosocial impact of this diagnosis.

**Methods** We conducted an observational study among symptomatic heterosexual men attending SHCs to evaluate the psychosocial impact of an NSGI diagnosis compared with a diagnosis of *Chlamydia trachomatis* (CT), *Neisseria gonorrhoeae* or no abnormalities detected focusing on the feasibility of our study methodology. Participants completed a computer-assisted self-interviewing (CASI) including two validated measures of psychosocial impact: the EQ-5D-5L health-related quality of life and Rosenberg Self-Esteem Scale, before diagnostic testing and 2 weeks after receiving test results (follow-up 1 (FU-1)) and a qualitative interview. We compared scores between diagnostic groups using paired t-tests, qualitative data were analysed thematically and feasibility was assessed by process analysis.

**Results** 60 men completed the baseline CASI (75% response rate). 46 (76.6%) were eligible for follow-up; 11/46 (23.9%) completed the follow-up CASI, and 3/11 (27.3%) completed the qualitative interview. 81.7% of all participants left CASI feedback at baseline: 73.5% reported the questionnaire as 'fine' or 'very good'. Qualitative interview participants reported the study was acceptable. Compared with baseline, among patients completing FU-1, only patients with a diagnosis of NSGI (p<0.05) or CT (p<0.05) showed increased EQ-5D-5L Index, whereas patients with a diagnosis of NSGI (p=0.05) showed decreased mean Rosenberg Self-Esteem Scale score.

**Conclusions** Although most participants indicated study acceptability at baseline, and we employed measures to increase retention (CASI questionnaires, reminder messages and a focus on men's health), we experienced high loss to follow-up. We found that heterosexual men attending SHCs with symptoms of urethritis experience both positive and negative psychosocial impacts

---

### Strengths and limitations of this study

► This study presents the first data investigating the psychosocial impacts of the common diagnosis of non-specific genital infection in heterosexual men attending sexual health clinics.
► This study used both validated scales and qualitative interviews to assess psychosocial impact.
► We used computer-assisted self-interviewing to collect data on psychosocial impact as this has been shown to improve data quality in sensitive topics.
► The study was small and was conducted in two London sexual health clinics, which may limit the generalisability of our results.
► There was high loss to follow-up after baseline, which may have biased our results.

following their clinic attendance, which warrants further investigation.

## INTRODUCTION

Non-specific genital infection (NSGI) or, non-*Chlamydia trachomatis* (non-CT), non-*Neisseria gonorrhoeae*-associated urethritis (NGU), is the third most frequently diagnosed condition in heterosexual men attending sexual health clinics (SHCs) in the UK, with 30 381 cases/year compared with *Chlamydia trachomatis* (CT) (42 483/year) and first presentation of genital warts (32 656/year).[1] NSGI is a diagnosis of exclusion; men with urethral symptoms are screened with urethral microscopy in SHCs, and if this demonstrates urethritis with no gonococci, a provisional diagnosis of NGU is made.[2] If urinary nucleic acid amplification tests (NAATs) sent for NG

and CT are both negative, the diagnosis is recorded as NSGI.

Although a common condition, NSGI remains poorly understood. NSGI can be caused by a variety of sexually transmitted infections (STIs), including *Mycoplasma genitalium* and *Trichomonas vaginalis*, which are not routinely tested for in SHCs, and many cases have an uncertain aetiology.[2] Microscopy, on which the diagnosis rests, has poor specificity for known causes of urethritis[3] and is subject to considerable observer variation.[4] National guidelines recommend that men diagnosed with NGU are informed they may have an STI, given same day empirical antibiotics, advised their partners should attend for STI testing and empirical treatment and to abstain from sex until treatment is completed.[2] Men usually receive CT/NG NAAT results within 8 days.[5] Following first-line antibiotics for NSGI, 10%–20% of men return because of persistent symptoms.[6 7]

Men who undergo STI testing and are diagnosed with an STI report negative impacts on their psychosocial well-being, including feelings of stigma[8–10]; however, to our knowledge, there are no published data on the psychosocial impact of NSGI diagnosis among men. Stigma has been shown in men to be associated with a delay in testing for STIs and decreased willingness to notify their casual partners of their STI diagnosis.[11] Studies on psychosocial impact, partner notification and quality of life impact have predominantly been carried out in women,[9 12–14] with few studies recruiting men. A greater understanding of the degree and nature of psychosocial impacts of a diagnosis of an STI, or a 'presumed STI' such as NSGI, in men, is important to optimise interventions, including partner notification and risk-behaviour modification, which aim to reduce the negative impact of STIs, such as complications of infection.

Men are reportedly poor users of health services, largely due to culturally dominant ideals of masculinity that equate illness with weakness, which may impede care seeking.[15–18] In sexual healthcare, there has traditionally been a focus on female reproductive health,[19] and this may further discourage male attendance.[19 20] However, men are willing to respond openly to questions about their sexual health, particularly when enquiries are focused specifically on men's needs.[19 21]

We decided to focus on heterosexual men in this study as there are important differences between men who have sex with men (MSM) and men who have sex exclusively with women (MSEW) in their access of SHC services, reported sexual health risks, their own perception of risk of STIs and sexual health and well-being outcomes,[22] suggesting that their experience of diagnosis of a genital infection may also be different. Due to these important differences between MSM and MSEW, we decided that psychosocial impacts of an STI diagnosis in MSM merited studying separately.

Because of a lack of models to measure the psychosocial impact of an NSGI diagnosis among men, and the potential difficulty in recruiting and retaining this group

in sexual health studies, we designed a small-scale longitudinal mixed-method feasibility study to inform on our research methods and better understand the psychosocial and health-related quality of life (HRQoL) status of symptomatic heterosexual men before and after receiving a diagnosis of NSGI, CT, NG or 'nothing abnormal detected' (NAD).

## METHODS

### Study design

We conducted an observational, longitudinal study in symptomatic heterosexual men using patient completed questionnaires assessing HRQoL and psychosocial status before and after receiving STI test results and individual qualitative interviews to evaluate experiences at the SHC and acceptability of study methods.

Eligibility criteria were: heterosexual men presenting with urethral symptoms (dysuria, discharge or urethral discomfort) and aged ≥16 years at either of two participating London SHCs between September and October 2016. We excluded men who were asymptomatic, not available for follow-up and who were unable to understand English.

Following informed consent, participants were invited to complete a baseline computer-assisted self-interviewing (CASI) assessing psychosocial and HRQoL status in clinic prior to consultation, a follow-up CASI after receipt of all test results 1–2 weeks after baseline (follow-up 1 (FU-1)) and a qualitative interview 2–4 weeks after baseline (follow-up 2 (FU-2)).

The participant pathway is detailed in figure 1. After completion of the baseline CASI, any participants diagnosed with an STI other than CT, NG or NSGI, or who did not have both urethral microscopy and urinary NAATs, were withdrawn from the study. Once participants were notified of their NAAT CT/NG results by the SHC (phone call/text message), the research team texted participants an electronic link to the FU-1 CASI.

CASIs were optimised for web-enabled devices including smartphones. We offered participants the choice to complete CASIs using their own device or a study tablet device and use of a private room in clinic. Study staff sent ≤2 text message reminders for each follow-up. Participants who did not complete FU-1 were considered ineligible for FU-2. Participants received gift voucher reimbursements: £5 (baseline), £10 (FU-1) and £20 (FU-2).

We categorised participants into four diagnostic groups based on microscopy and NAAT results: NSGI (NGU diagnosis on urethral microscopy and CT/NG NAATs negative); NG (NG on urethral microscopy and/or positive NAAT); CT (positive NAAT); and NAD (no urethritis on microscopy and CT/NG NAATs negative).

### Questionnaire design

At both baseline and FU-1 CASIs, the EQ-5D-5L, a generic measure of HRQoL,[23] and the Rosenberg Self-Esteem

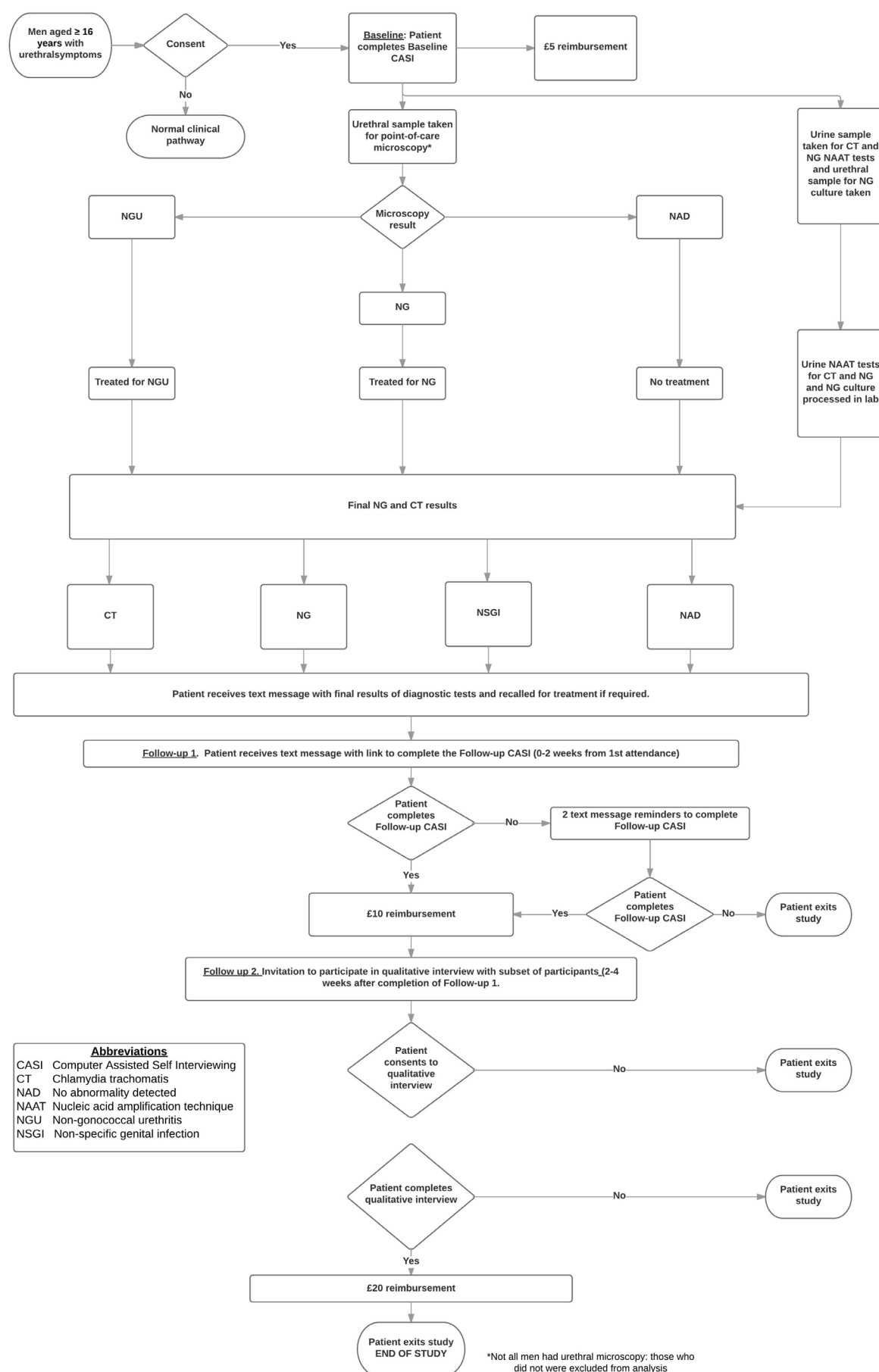

**Figure 1** Patient pathway. NG, *Neisseria gonorrhoeae*.

Scale (RSES),[24] a measure of potential stigmatising effects of screening for STIs, were used. The EQ-5D-5L is the preferred utility measure recommended by the UK National Institute for Health and Clinical Excellence[22] and has five dimensions: mobility, self-care, usual activities, pain/discomfort and anxiety/depression. Weighted UK preference values are linked to the self-reported health state scores for a 0–1 Index value, where 0 is death and 1 is perfect health. The RSES is widely used in social science research to measure self esteem; it uses a scale of 0–30, where a score less than 15 may indicate low self esteem, 15–25 normal self-esteem and 25–30 high self-esteem.

In addition, the baseline CASI included the Multidimensional Scale of Perceived Social Support (MSPSS)[25] to assess coping resources. The MSPSS is widely used to assess perceived social support and consists of three domains: family, friends and significant others. Mean MSPSS scores from 1 to 2.9 may be considered low support; 3–5 moderate support and 5.1–7 high support.[26]

We collected data on sociodemographics in the baseline CASI and current relationship status in both the baseline and FU-1 CASI. In the FU-1 CASI, we included a panel of questions previously developed to assess impact of CT diagnosis in women,[13] which we modified for use in men by removing a pregnancy-related question. Participants were asked to indicate acceptability of the CASIs (free-text response) at baseline and FU-1 (online supplementary appendices 1–3).

## Patient and public involvement

There was no formal patient and public involvement informing the research question or study procedures. Clinician experiences with patients diagnosed with NSGI informed the need for investigation of the potential harms resulting from this diagnosis. The results have been disseminated via the study groups' community advisory group (formed after the research was planned and implemented).

## Sample size

As this was a feasibility study with no precedent, no formal sample size could be calculated. We planned to approach 100 patients at baseline as a convenience target, and assuming approximately 50% refusal, we expected 50–60 patients to be enrolled and estimated ~50% attrition at each follow-up.

## Outcome measures
### Primary outcome

Qualitative and quantitative process data on recruitment and retention rates at baseline and follow-up to inform on feasibility of study design. Evaluation of potential relevance of questions validated in other populations for men diagnosed with NSGI.

### Secondary outcomes

Percentage change from baseline to FU-1 in mean EQ-5D-5L Index value, and mean RSES value in men diagnosed with CT, NG, NSGI or NAD.

We did not have prior expectations about different impacts of these diagnoses, and as there are no studies comparing the impact of CT to NG or NSGI in men, this was exploratory. We were interested in investigating whether NSGI was perceived as a benign diagnosis or whether it elicited negative psychosocial impacts that have been reported in men diagnosed with STIs such as CT.[10]

## Data analysis
### Questionnaire data

For all eligible men, RSES score, MSPSS score and EQ-5D-5L Index value were calculated according to published methods.[23–25 27]

For men in each diagnostic category, paired t-tests were used to compare mean values for the EQ-5D-5L Index, RSES and MSPSS at baseline and to compare differences in mean values for EQ-5D-5L Index and RSES score, from baseline to follow-up.

To test for non-response bias, we compared sociodemographic data of those who completed both baseline and follow-up to those who were lost to follow-up (LTFU), using Pearson's $\chi^2$ tests and Kruskal-Wallis tests. Statistical significance was set at $p<0.05$. Statistical analysis was carried out using STATA V.13.1.

### Qualitative data

Audio recordings were transcribed, and transcripts were checked for accuracy against the recording. Cleaned transcripts were then imported into NVivo (V.10, 2012) for analysis. Two researchers assigned themes to text independently, with discrepancies resolved by consensus before reporting final themes. All names of participants presented here are pseudonyms.

## RESULTS

We approached 80 patients, of whom 60 (75%) enrolled and completed the baseline CASI. Forty-six (76.6%) were eligible for follow-up; of these, 11 (23.9%) completed the FU-1 CASI, and 3 of these (27.3%) completed FU-2 (figure 2). Final diagnoses among all eligible participants (n=46) were: CT: 7 (15.2%); NG: 2 (4.3%); NSGI: 23 (50%); and NAD: 14 (30.4%), and among participants who completed FU-1 (n=11) were: CT: 4 (36.4%); NSGI: 4 (36.4%); and NAD: 3 (27.3%). None of the participants diagnosed with NG completed FU-1. Two participants with NSGI and one with NAD completed FU-2. Clinical records indicated that all participants were notified of their CT/NG NAAT diagnosis within 7 days of completing baseline.

Men who reported urethral symptoms but had no abnormalities on urethral sample testing and negative CT and NG results (categorised as NAD) had a variety of non-STI causes for their symptoms diagnosed, including irritant or *Candida* balanitis, lower urinary tract symptoms, which were likely prostatic in origin, non-infectious genital pain syndromes and symptoms secondary to urethral trauma.

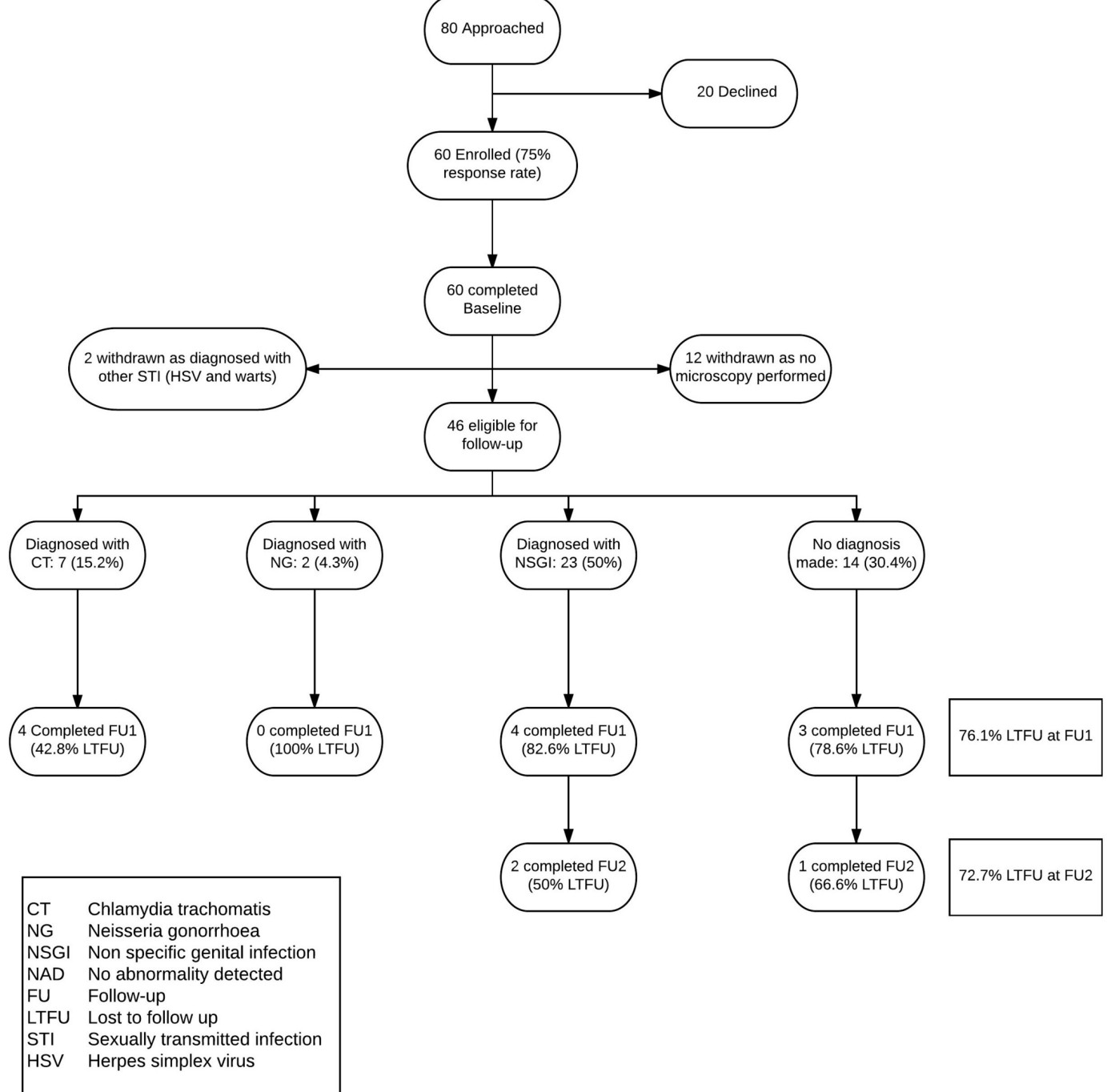

CT      Chlamydia trachomatis
NG      Neisseria gonorrhoea
NSGI    Non specific genital infection
NAD     No abnormality detected
FU      Follow-up
LTFU    Lost to follow up
STI     Sexually transmitted infection
HSV     Herpes simplex virus

**Figure 2** Enrolment and follow-up chart.

Participants' sociodemographic factors at baseline are presented in table 1. Most were between 25 and 35 years old (60.9%), and over half reported their ethnicity as non-white (52.2%). A percentage of 80.4 indicated they were not in a relationship, and many had previously been diagnosed with STIs (30.4% NGU, 28.3% CT, 13.0% genital herpes or warts and 8.7% NG).

Pearson's $\chi^2$ tests and Kruskal-Wallis test results for non-response bias were not statistically significant so we assumed data were missing at random. We compared participants who completed FU-1 to those LTFU by employment, age, education, ethnicity and current relationship status; none of these comparisons showed significant differences.

### Process evaluation

All patients who completed the baseline CASI did so using the study tablet device in a private room in the clinic. Participants completed the baseline CASI in about 5 min. All participants completed the follow-up CASI at home on a participant-owned device.

CASI feedback suggested good acceptability: of the total 60 who completed the baseline CASI, 81.7% left feedback, of whom 73.5% reported that the questionnaire was

**Table 1** Sociodemographic characteristics of enrolled participants at Baseline by final diagnosis

| | CT n (%) | NG n (%) | NSGI n (%) | NAD n (%) | Total n |
|---|---|---|---|---|---|
| **Age (year)** | | | | | |
| <24 | 5 (71.43) | 1 (50) | 5 (21.74) | 3 (21.43) | 14 |
| 25–35 | 2 (28.57) | 1 (50) | 17 (73.91) | 8 (57.14) | 28 |
| >35 | 0 (0) | 0 (0) | 1 (4.35) | 3 (21.43) | 4 |
| **Ethnicity** | | | | | |
| White | 5 (71.43) | 0 (0) | 10 (43.48) | 6 (42.86) | 21 |
| Asian | 0 (0) | 1 (50) | 2 (8.7) | 2 (14.29) | 5 |
| Black | 1 (14.29) | 1 (50) | 10 (43.48) | 2 (14.29) | 14 |
| Other | 1 (14.29) | 0 (0) | 1 (4.35) | 3 (21.43) | 5 |
| Prefer not to say | 0 (0) | 0 (0) | 0 (0) | 1 (7.14) | 1 |
| **Age at leaving full-time education** | | | | | |
| 16 years or less | 0 (0) | 0 (0) | 1 (4.35) | 3 (21.43) | 4 |
| 17 or 18 years | 0 (0) | 0 (0) | 6 (26.09) | 0 (0) | 6 |
| 19 years or over | 4 (57.14) | 0 (0) | 14 (60.87) | 10 (71.43) | 28 |
| Still in full-time education | 3 (42.86) | 2 (100) | 2 (8.7) | 1 (7.14) | 8 |
| **Highest educational achievement** | | | | | |
| No qualifications/ no formal qualifications | 1 (14.29) | 0 (0) | 1 (4.35) | 0 (0) | 2 |
| 1–5 General Certificates of Secondary Education (GCSEs) | 0 (0) | 0 (0) | 3 (13.04) | 2 (14.29) | 5 |
| Two or more A-levels or equivalent | 1 (14.29) | 1 (50) | 4 (17.39) | 2 (14.29) | 8 |
| Bachelor's degree or equivalent and higher | 5 (71.43) | 1 (50) | 13 (56.52) | 8 (57.14) | 27 |
| Other | 0 (0) | 0 (0) | 2 (8.7) | 2 (14.29) | 4 |
| **Do you have any long-standing condition?** | | | | | |
| Yes | 1 (14.29) | 0 (0) | 1 (4.35) | 2 (14.29) | 4 |
| No | 6 (85.71) | 2 (100) | 22 (95.65) | 12 (85.71) | 42 |
| **Previously diagnosed with CT?** | | | | | |
| Yes | 4 (57.14) | 1 (50) | 7 (30.43) | 1 (7.14) | 13 |
| No | 3 (42.86) | 1 (50) | 16 (69.57) | 13 (92.86) | 33 |
| **Previously diagnosed with NG?** | | | | | |
| Yes | 0 (0) | 2 (100) | 2 (8.7) | 0 (0) | 4 |
| No | 7 (100) | 0 (0) | 21 (91.3) | 14 (100) | 42 |
| **Previously diagnosed with NGU?** | | | | | |
| Yes | 2 (28.57) | 0 (0) | 9 (39.13) | 3 (21.43) | 14 |
| No | 5 (71.43) | 2 (100) | 14 (60.87) | 11 (78.57) | 32 |
| **Previously diagnosed with herpes/warts?** | | | | | |
| Yes | 3 (42.86) | 0 (0) | 1 (4.35) | 2 (14.29) | 6 |
| No | 4 (57.14) | 2 (100) | 22 (95.65) | 12 (85.71) | 40 |
| **Relationship status** | | | | | |

Continued

**Table 1** Continued

| | CT n (%) | NG n (%) | NSGI n (%) | NAD n (%) | Total n |
|---|---|---|---|---|---|
| Married and living with wife | 0 (0) | 0 (0) | 1 (4.35) | 2 (14.29) | 3 |
| Cohabiting but not married | 1 (14.29) | 0 (0) | 2 (8.7) | 2 (14.29) | 5 |
| Separated, divorced or widowed | 0 (0) | 0 (0) | 1 (4.35) | 0 (0) | 1 |
| Single (ie, never married) | 6 (85.71) | 2 (100) | 19 (82.61) | 9 (64.29) | 36 |
| Prefer not to say | 0 (0) | 0 (0) | 0 (0) | 1 (7.14) | 1 |
| Break up in the last 30 days | | | | | |
| Yes | 1 (14.29) | 0 (0) | 3 (13.04) | 5 (35.71) | 9 |
| No | 6 (85.71) | 2 (100) | 20 (86.96) | 9 (64.29) | 37 |

CT, *Chlamydia trachomatis*; NAD, nothing abnormal detected; NG, *Neisseria gonorrhoeae*; NGU, non-*Neisseria gonorrhoeae* (non-NG)-associated urethritis; NSGI, non-specific genital infection.

'fine' or 'very good' in a free-text box at the end of the questionnaire.

Participants suggested improvements to the study during qualitative interviews: to use the research as a tool for health promotion and to make CASI questions more clearly relevant to sexual health. Insight into LTFU was provided by one participant, who said he might not have continued with the study if he had received a positive STI diagnosis:

> I reckon if my tests had not come back the way I wanted them to, I probably wouldn't have done this last [follow-up]. … I might've still said, 'Yes, let's do it.' I don't know but I… Just sitting here, thinking about it now, I probably wouldn't complete the rest of it. (Lucas, NAD diagnosis)

### CASI analysis

Distributions of participants' psychosocial scores at baseline by final diagnosis are shown in table 2. At baseline (table 2A) participants ultimately diagnosed with NG had significantly lower ($p \leq 0.050$) mean MSPSS scores (indicating lower social support), compared with men ultimately diagnosed with NSGI or NAD, as well as lower scores as compared with those diagnosed with CT ($p=0.059$). Participants with NG also had lower mean RSES scores (indicating lower self-esteem) at baseline as compared with those with NSGI ($p=0.035$). EQ-5D-5L Index value at baseline did not vary significantly between diagnosis groups.

We found men with NSGI and CT showed a significant increase in mean EQ-5D-5L Index value (indicating better HRQoL) between baseline and FU-1 (+17% ([$p=0.006$) and +219% ($p=0.005$), respectively) (table 2B). There was no change in mean EQ-5D-5L Index value for men diagnosed with NAD. In contrast, we found decreases

in mean RSES scores from baseline to FU-1 in men with NSGI (−30%, $p=0.05$).

Participants who reported that they had been diagnosed with an infection (n=5, four with a final diagnosis of CT and one with NSGI) completed a panel of questions in the FU-1 CASI to assess their feelings about their diagnosis. All participants reported that they were 'not worried as it is curable' and did not think the diagnosis would change how their friends thought about them, but only one participant (diagnosed with CT) reported he had talked to his friends about his diagnosis. All participants reported concern that they might get the infection again, and the majority of those with a CT diagnosis (75%) reported concern about exposure to other STIs, feeling 'dirty' and embarrassed, being scared to tell their partners about their diagnosis and worried that their partners thought they had been unfaithful.

### Qualitative analysis

In interviews, participants compared different STIs as a way to contextualise their level of concern about their own diagnosis. For one participant, his main concern was the possibility of being diagnosed with HIV:

> Look, to be honest with you, the key one everyone really wants to know about is HIV … all the other ones are treatable quite easily… (Lucas, NAD diagnosis)

There were several points within the interviews where participants expressed concern at being diagnosed with an infection they did not know about, indicating that non-specific diagnoses may cause negative impact:

> …to be honest, I'm a little bit worried about this because there is no name of the new bacteria, so we don't know if it's dangerous or not… (Liam, NSGI diagnosis)

**Table 2** Distribution of psychosocial scores by final diagnosis

**(A) Distribution of psychosocial scores within the baseline group and results from Student's t-tests**

| | | Score | 95% CI | P values NSGI versus CT | P values NSGI versus NAD | P values CT versus NAD | P values NG versus CT | P values NG versus NSGI | P values NG versus NAD |
|---|---|---|---|---|---|---|---|---|---|
| EQ-ED-5L (index value) | NG | 0.46 | 0.28 to 0.64 | 0.238 | 0.367 | 0.708 | 0.700 | 0.794 | 0.882 |
| | NSGI | 0.59 | 0.46 to 0.72 | | | | | | |
| | CT | 0.44 | 0.19 to 0.69 | | | | | | |
| | NAD | 0.49 | 0.32 to 0.67 | | | | | | |
| Rosenberg Self-Esteem Scale | NG | 15.5 | 0.0 to 72.7 | 0.358 | 0.816 | 0.354 | 0.09 | 0.035 | 0.071 |
| | NSGI | 22.0 | 20.0 to 24.0 | | | | | | |
| | CT | 20.4 | 17.5 to 23.4 | | | | | | |
| | NAD | 22.4 | 19.6 to 25.6 | | | | | | |
| Multidimensional Scale of Perceived Social Support | NG | 2.5 | 2.0 to 3.0 | 0.583 | 0.610 | 0.918 | 0.059 | 0.016 | 0.040 |
| | NSGI | 3.1 | 2.9 to 3.2 | | | | | | |
| | CT | 3.2 | 2.8 to 3.6 | | | | | | |
| | NAD | 3.2 | 2.9 to 3.4 | | | | | | |

**(B) Distribution of psychosocial scores by final diagnosis within the FU group and restricted to those with an FU record comparing baseline and FU scores**

| | | Baseline* | | FU-1 | | % difference of score Baseline* and FU-1 | P values Baseline* versus FU-1 |
|---|---|---|---|---|---|---|---|
| | | Score | 95% CI | Score | 95% CI | | |
| EQ-ED-5L (index value) | NSGI | 0.82 | 0.79 to 0.86 | 0.96 | 0.85 to 1.07 | 17 | 0.006 |
| | CT | 0.27 | 0.14 to 0.39 | 0.85 | 0.56 to 1.14 | 219 | 0.005(MF1) |
| | NAD | 0.59 | 0.16 to 1.00 | 0.75 | 0.25 to 1.24 | 27 | 0.658 |
| Rosenberg Self-Esteem Scale | NSGI | 22.5 | 14.6 to 30.3 | 15.7 | 7.81 to 23.69 | −30 | 0.050 |
| | CT | 20.7 | 12.81 to 28.69 | 20.0 | 5.71 to 34.29 | −3 | 0.444 |
| | NAD | 22.7 | 8.11 to 37.22 | 19.3 | 15.5 to 23.1 | −15 | 0.197 |

*Restricted to patients who completed FU-1: no patients diagnosed with NG were present at FU-1.
CT, *Chlamydia trachomatis*; FU, follow-up; NAD, no abnormality detected; NG, *Neisseria gonorrhoeae*; NSGI, non-specific genital infection.

Among the two interview participants diagnosed with NSGI, one described anxiety following his presumptive CT diagnosis in clinic. He was in a long-term monogamous relationship and so was not expecting an STI diagnosis:

My heart's racing. I'm thinking, 'but how, I can't.' … I just had a lot of thoughts in my head. … And I told [the doctor], 'I can't. I don't know how I would have this.' He's saying, 'Well, whatever you have, it's sexually transmitted.' I said, 'But it can't be.' So of course I start getting things in my head. (Jacob, NSGI diagnosis) (online supplementary appendix 4)

## DISCUSSION

In this small observational feasibility study, we assessed patient reported self-esteem and HRQoL using validated scales in heterosexual men presenting to SHCs with urethral symptoms, before undergoing STI testing and after receiving all test results. This is the first study to investigate the HRQoL and psychosocial impact of a curable STI diagnosis in symptomatic men attending SHCs, using validated scales, and is one of the few available with data on STIs and EQ-5D-5L.[28]

We found increased EQ-5D-5L Index value (indicating increased HRQoL) in men diagnosed with NSGI or CT from baseline to FU-1. Published studies have reported that some men diagnosed with CT reported a lack of concern regarding the diagnosis as they perceive CT as a relatively minor infection.[8 10 14 29] This was supported by one of our qualitative interview participants. The increases we found in the EQ-5D-5L Index value may reflect decreases in the anxiety and pain/discomfort domains of the scale from just prior to consultation at baseline to receipt of all test results in FU-1. Shoveller et al[30] reported that the majority of men diagnosed with an STI in SHCs had reported feeling anxious waiting for potentially bad news, and so this finding may reflect that men found relief following treatment and explanation of their symptoms and receiving negative results for more serious infections such as HIV. Due to our small sample size, we were unable to find significant associations between participant satisfaction with their clinic visit and relief from symptoms and infection status in our CASI FU-1 data.

Furthermore, in men diagnosed with NSGI, we found decreased mean RSES score (indicating decreased self-esteem) from baseline to approximately 1 week after receiving all STI test results. These findings support data from previous qualitative studies where men diagnosed with an STI in an SHC reported negative psychosocial impacts including stigma, anxiety, shame, isolation, concerns regarding relationships, a loss of social status, vulnerability, a lack of privacy and fear of STI testing, particularly urethral swab testing.[8 10 30–32] We did not find a decrease in self-esteem in men diagnosed with CT, which might suggest that NSGI is perceived differently to

CT. In our qualitative interviews, participants with NSGI reported concerns regarding the uncertainty of the diagnosis and fear of the impact of a possible STI diagnosis on their relationships; these factors may be important in the observed decreased self-esteem in these men.

### Limitations

Our choice of HRQoL tool, the EQ-5D-5L, may not be sensitive enough to detect impacts caused by the diagnosis of CT, NG or NSGI, which are unlikely to cause problems in at least two of the five domains: mobility and self-care. Studies investigating HRQoL and sexual health have found few significant differences with comparator groups using generic QoL instruments alone.[33] STI-specific HRQoL and psychosocial impact tools have been developed for genital herpes and genital warts,[34–36] and several studies have now also combined both EQ-5D-5L and STI-specific tools.[37–40]

Our study did not include asymptomatic men; hence, we cannot exclude the possibility that the negative impact on self-esteem observed with men was not related to being symptomatic or undergoing genital examination and a urethral swab.[10 30 32] We suggest that future studies consider including both symptomatic and asymptomatic men to better evaluate the impact of the experience of having symptoms, genital examination and invasive testing on psychosocial well-being and self-esteem.

Although the RSES and the EQ-5D-5L scales we used in this study have been validated in men and women, the panel of questions used to assess men's feelings about a diagnosis of CT or NSGI in the FU-1 CASI has not been validated in men or for non-CT STI diagnoses. In addition, we assessed psychosocial and HRQoL impact at a single time point, which does not measure durability of the observed decrease in self-esteem in this population, and data suggest psychosocial impact of STI diagnoses may decrease over time.[41]

We received feedback from our participants that some of the wording of questions on our CASIs did not appear to be directly relevant to our study aims. Future studies should ensure that data collection measures use terms that are clear and familiar to participants, possibly through development of questionnaires with focus groups. Some participants requested that future studies offer research participants material on health promotion, that is, material on the infection they were diagnosed with and how to reduce risk of infection.

Despite concerns that the target group of heterosexual men attending SHCs would be difficult to recruit,[42] we found 75% of men were happy to enrol in this study. This should encourage researchers to investigate views from this group regarding STI diagnosis and screening. A study investigating the psychosocial experiences of patients attending a clinic, which recruited men and women diagnosed with an STI from an SHC, reported a recruitment rate of 33%[31] and involved 10 interviews lasting between 20 min and 45 min each. Our higher baseline recruitment rate may be due to the use of CASI interviewing, fewer

interviews or because patients were recruited before they had any diagnosis made.

One of the difficulties associated with conducting longitudinal studies is attrition. In our study, of those eligible for follow-up, 24% overall completed the follow-up CASI, 25% of those with a diagnosis of NSGI, CT or NG and 21% of those with NAD. A longitudinal questionnaire study investigating community screening for CT reported follow-up rates of over 60%; however, eligibility for follow-up excluded those with positive STI test results, and 60% of the sample was female, reflecting the higher response rate among women.[12] Our study's qualitative component had a follow-up rate of 27% overall, 25% in those who had any positive diagnosis and 33% in those with NAD. By comparison, a study including men, investigating patient views on CT screening recruited 6.6% of those who completed the baseline questionnaire to a qualitative interview, which was below their target of 10%.[43]

We embedded several mechanisms in our study to improve participant acceptability and retention. Privacy when completing sensitive questions has been shown to improve participant acceptability for revealing potentially stigmatising information[44]; we used a web-based CASI and the option for participants to self-complete the questionnaire in a private room in the clinic. Potential research participants, and men specifically, are more likely to participate in research that represents their concerns[19 21 45]; eligibility for this study was restricted to those patients who were likely to be diagnosed with an STI and focused on the impact of those diagnoses. In addition, reimbursements have been shown to play a part in increasing enrolment and retention in research[45 46]; we reimbursed participants in increasing amounts to reflect the time and effort we estimated were necessary to complete the research. Despite these mechanisms, and positive participant feedback on the baseline CASI and in our qualitative interviews, we experienced high LTFU. High LTFU resulted in low numbers of respondents within diagnostic categories, which meant we were unable to adjust our analysis for common confounders, such as education and socioeconomic status, known to be associated with gonorrhoea.[47 48] This may be a source of bias in our results. We did not find significant differences between groups in terms of non-response bias and other characteristics; however, this is not surprising as the power to detect differences between groups is greatly affected by sample size, and our study included a small number of participants. Recruiting larger numbers of participants in future studies will increase the statistical power to detect any important differences between groups.

We explored potential correlates and causes for LTFU within our dataset. Men's socioeconomic status and age have been seen to affect both their knowledge of and interaction with health services.[20] However, we did not find significant differences in these factors between those LTFU and those who completed FU-1. One participant indicated in qualitative interview that if he had received a positive STI diagnosis, it was unlikely that he would

have agreed to continue with the research. This may give insight into why no patients receiving a positive CT or NG diagnosis agreed to FU-2.

We had planned to further explore study acceptability by inviting participants who failed to complete FU-1 to qualitative interviews; however, we followed research ethics committee recommendations to consider that these participants had passively withdrawn from the study and refrained from inviting them to FU-2. Yet, qualitative and quantitative methods of data collection are viewed differently by respondents.[49 50] One participant described the difference between his experiences completing the survey and participating in an interview:

> …when I'm sitting here with you, obviously it's just me and you and you've got my hundred percent full attention, haven't you? But then when I'm sitting at my desk at work, trying to complete a survey and there's other things going on and people asking me questions and 'can you do this, can do you that'; it's different. (Lucas, NAD diagnosis)

Although we cannot know if participants who did not complete FU-1 would have participated in a qualitative interview if they had been invited, one participant left feedback on the baseline CASI indicating he would have preferred to elaborate on his answers, and several participants informally conveyed this to the researcher at baseline. Research has shown that one unique benefit of qualitative interviews is that it provides participants with an opportunity to expand or contextualise issues.[49 50] In addition, the insights we gained from the limited qualitative interviews we conducted indicate the potential usefulness of this method to understand nuances of acceptability that may lead to reduced LTFU. We therefore recommend that future feasibility studies include qualitative components as a means to allow participants, including those who fail to complete survey questionnaires, to provide important feedback on study methods. Future studies could also consider offering a qualitative interview at the point of recruitment, not only at follow-up, to widen the possibility of a qualitative exploration among participants who do not find the CASI questionnaire acceptable.

The issue of gender concordance between participants and interviewers/researchers has been discussed in the literature,[51] and the fact that our study recruiter was female may have affected recruitment and retention. A study investigating attitudes to Viagra in heterosexual Hispanic men, using telephone qualitative interviews, found that changing from female to male recruiters and data collectors increased their recruitment.[42] Participants in our study were offered the option of a male or female interviewer but may not have felt comfortable with asking for a particular interviewer and so this may not have been sufficient to reduce the potential impact of gender discordance on acceptance of the invitation to an interview in our study. We suggest future studies involving qualitative interviews should consider the gender of recruiters and interviewers.

## CONCLUSIONS

Our study found that heterosexual men with urethral symptoms may experience a range of psychosocial and HRQoL impacts following SHC attendance and STI testing. More work is needed to investigate the most appropriate methodology for investigating sexual health and QoL, particularly around choosing between condition specific and generic measures, and reducing loss to follow-up in longitudinal studies. Future research in this area is needed to test the generalisability of our findings as to whether there are significant psychosocial harms of giving presumptive and non-specific diagnoses.

**Acknowledgements** Thank you to the participants who took part in the study, our collaborators and the steering committee for their advice, including: Atlas Genetics, who were awarded the Innovate UK SBRI grant; Aquarius Population Health; Scientific Steering Committee (Chris Price, Kate Folkard and David Livermore) and the clinical leads at participating clinics (Richard Lau and Paul Lister).

**Contributors** STS conceived of the Study. RH-T, SSF and EMH-E initiated the study design. AP, SSF, RH-T and EMH-E implemented the study. MF and EMH-E provided statistical oversight. RH-T wrote the initial draft of the manuscript. All authors contributed to and approved the final manuscript.

**Funding** This study was funded by an Innovate UK (SBRI grant no. 971452) awarded to Atlas Genetics Ltd.

**Disclaimer** Neither Atlas Genetics nor Innovate UK had a role in the design, collection, management, analysis, interpretation of data or decision to submit a report. The views expressed are those of the authors and not necessarily those of the NIHR, the NHS or the Department of Health.

**Competing interests** The Applied Diagnostic Research and Evaluation Unit at St George's, University of London (STS, EMH-E, SF and AP) receives funding from the National Institute of Health Research (NIHR) i4i Programme (grant number II-LB-0214-20005), Atlas Genetics, Alere, Hologic Cepheid, SpeeDx, Sekisui and Becton Dickinson to develop Point of Care Tests for STIs.

**Patient consent** Obtained.

**Ethics approval** This study was approved by the London Bridge Research Ethics Committee (REC reference 16/LO/0955).

**Provenance and peer review** Not commissioned; externally peer reviewed.

**Data sharing statement** Due to ethical concerns regarding the sensitive nature of the research, transcripts of qualitative interviews are not available for data sharing. All other data supporting this study are provided as supplementary information accompanying this paper.

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
