## [Reviewer comments · BMJ Open]

ARTICLE DETAILS

TITLE (PROVISIONAL)	Health related quality of life and psychosocial impacts of a diagnosis of non-specific genital infection in symptomatic heterosexual men attending UK Sexual Health clinics: a feasibility study.
AUTHORS	Hill-Tout, Rachel; Harding- Esch, Emma; Pacho, Agata; Furegato, Martina; Fuller, Sebastian; Sadiq, S. Tariq

VERSION 1 – REVIEW

REVIEWER	Sarah Burkill Karolinska Institutet, Sweden
REVIEW RETURNED	02-Aug-2017

GENERAL COMMENTS	This study considered the feasibility of conducting research into the impact of a non-specific genital infection diagnosis on the wellbeing of patients through considering health related quality of life indicators. A mixed methods approach was taken. Overall a nicely done paper which highlights the importance of studying the often neglected issues surrounding the psychosocial impact of an STI diagnosis, in this case specifically NSGI's. Some comments for consideration below: Introduction 1- The justification for only studying heterosexual men was not specified. Inclusion of why only this group were considered would bring clarity. Would different effects of an NSGI diagnosis be expected amongst those with other sexual identities? Otherwise a strong introduction which clearly specifies the aim of the study. Methods 1- Have previous studies looked at the impact attendance at an STI clinic has in a more general sense, including asymptomatic men with negative results? If so it would strengthen the paper for these studies to be discussed to justify not including asymptomatic men. Without asymptomatic men, in particular asymptomatic men with no subsequent positive results, the impact on men attending in general is not known, so it is difficult to infer whether the impact on those with symptoms is due to attendance per se, or due to presenting with symptoms. 2- Some men who were symptomatic (which by definition they had to be to meet the inclusion criteria) did not ultimately have any infection. It would be helpful at some point in the paper to briefly discuss alternative reasons for why these men had symptoms to clarify that there are other possible reasons for them. 3- T-tests to compare differences between groups both in terms of non-response bias, and the distribution of other characteristics were not significant, but this is not particularly surprising given the small
--

	sample size and subsequent low power- mentioning this as a limitation and discussing how power to detect differences between groups will increase if larger numbers are recruited in future studies would be beneficial. 4- Continuing from the point above on power- It would be my suspicion that participants with positive diagnoses would be less likely to continue with the study (as you state in your qualitative work)- as far as I can tell all diagnoses have been compared individually in a t-test to NAD. Combining all diagnoses into one group and comparing to NAD in one t-test may provide slightly more power to detect whether those with a positive diagnosis of any kind are less likely to continue with the study, which may highlight differences in response bias. The influence may of course be different depending on the diagnosis, but this could help in obtaining an overall picture. Discussion 1- NAD patients also showed decreased self-esteem scores, which shows the possible importance of including asymptomatic men. It may be the clinic visit itself which is having an impact. A reference to past studies looking at the overall impact of a clinic visit on psychosocial/esteem outcomes with a brief discussion would be beneficial. 2- Are you able to distinguish between those who answered the questions at each time point at the clinic, and who at home? Being at home, or otherwise using a device owned by the participant to answer the questions in a place of their choosing is thought to increase the likelihood of respondents reporting sensitive information. Looking at whether there is greater disclosure amongst those who did not answer in the clinic could provide valuable insight into how best to collect this information in the future. Given that this is a feasibility study which could influence the implementation of future research, this could be especially important. Other comments It should be specified more often that it is heterosexual men to whom these results apply (e.g. in the abstract article summary, the first bullet point states this is a common diagnosis amongst men in general. If it is also common amongst men with other sexual identities further justification for selecting only heterosexual men should be given). Table 1- I don't think the % are necessary with such small numbers. If you want to include them, it would be better to include in parentheses next to the n, so n(%). Otherwise the table is twice the width without adding any new information. Table 2- Which distribution has been used to calculate the p-values? For sample sizes under 30, students t-tests are conventionally used- have they been used here? The STROBE form specifies inclusion of study design in the title or early on in the paper- I would suggest including the term 'observational study' somewhere in the abstract or early on in the paper to clarify the design. Overall a very nice piece of work
--	---

REVIEWER	Pythia Nieuwkerk Academic Medical Center, Amsterdam, the Netherlands
REVIEW RETURNED	22-Aug-2017

GENERAL COMMENTS	This study investigated the health-related quality of life and psychosocial impact of a diagnosis of non-specific genital infection in
--

	symptomatic men attending sexual health clinics in a small scale longitudinal mixed-method feasibility study. The authors found that loss to follow up was high. Although most men found the study acceptable at the baseline measurement before diagnosis, less than 25% completed the follow-up measurement after diagnosis. Men who completed both the baseline and follow-up measurement reported significant decreases in their health related quality of life and self-esteem after diagnosis. I wonder if the authors could explain a bit more about the rationale of the study. The authors mention in the introduction that there are no published data on the psychosocial impact of non-specific genital infection diagnosis among men. But what would be the relevance of having information about the psychosocial impact? In many medical conditions, a low (mental) health related quality of life is related with a poor retention in care and a poor adherence to medication recommendations including treatment adherence. Could that be of any relevance? Given that these men receive recommendations about taking antibiotics, inform their partners and to abstain from sex until treatment is completed. The authors compared health-related quality of life and self-esteem between men who received a diagnosis of non-specific genital infection, chlamydia trachomatis, Neisseria gonorrhoeae and no-abnormality detected. Did the authors have a-priori expectations about different impacts of receiving these diagnoses? Did they expect that some diagnosis would have more impact than others? Please explain. It was unclear to me how the scores for the EQ-5D-5L were analyzed because the authors presented a single mean score for this questionnaire. The EQ-5D-5L consist of 5 dimensions of quality of life that are scored on a 5-point Likert scale and a visual analogue scale. A utility score can be calculated on the basis of the scores on the 5 dimensions. Where does the mean EQ-5D score refer to? One of the dimensions, the VAS, the utility score or something else? Please explain. In the introduction section, the authors mention that one of the aims of this feasibility study was to inform their study methods. Despite the authors made many efforts to facilitate retention in the study, including giving financial incentives to participants, what would they do different (if anything) in a next study to prevent attrition? What advice would they give to future researchers aiming to investigate a similar group of subjects?
--	---

VERSION 1 – AUTHOR RESPONSE

Reviewer 1(R1)

R1Q1) The justification for only studying heterosexual men was not specified. Inclusion of why only this group were considered would bring clarity. Would different effects of an NSGI diagnosis be expected amongst those with other sexual identities? Otherwise a strong introduction which clearly specifies the aim of the study.

We have addressed this in the Introduction (page 3, lines 36-38 and page 4, lines 1-6)

“We decided to focus on heterosexual men in this study as there are important differences between men who have sex with men (MSM) and men who have sex exclusively with women (MSEW) in their access of SHC services, reported sexual health risks, their own perception of risk of STIs, and sexual health and well being outcomes, suggesting that their experience of diagnosis of a genital infection may also be different. The Natsal-3 study reported that in comparison with MSEW, MSM are more likely to report higher numbers of partners, higher rates of condomless sex, to perceive themselves at higher rates of STIs, a greater likelihood to attend SHCs, and higher rates of depression, substance use and non-volitional sex [1]. Due to these important differences between MSM and MSEW, we decided that psychosocial impacts of an STI diagnosis in MSM merited studying separately.”

R1Q2) Have previous studies looked at the impact attendance at an STI clinic has in a more general sense, including asymptomatic men with negative results? If so it would strengthen the paper for these studies to be discussed to justify not including asymptomatic men. Without asymptomatic men, in particular asymptomatic men with no subsequent positive results, the impact on men attending in general is not known, so it is difficult to infer whether the impact on those with symptoms is due to attendance per se, or due to presenting with symptoms.

We addressed this in the Discussion (page 10, lines 32-38 and page 11, lines 1-15)

"A weakness of our study is that we did not include asymptomatic men, hence we cannot exclude the possibility that the negative psychosocial impacts observed with men diagnosed with CT or NSGI were not related to the confounding factors of being symptomatic or the attendance at an SHC (including undergoing genital examination, and a urethral swab). Few studies have examined the impact of an SHC clinic visit on psychosocial and/or self-esteem outcomes for men, but those that have been conducted did report negative impacts; studies among heterosexual men attending SHCs who are diagnosed with an STI reported stigma, anxiety, isolation, concerns regarding relationships, feeling 'contaminated' and fear of exposure, as well as reluctance to attend an SHC clinic for help [2,3]. A focus group of young male adolescents reported that accessing SHCs was stressful due to fear of stigma and a loss of social status, shame, and embarrassment, a lack of privacy/confidentiality, and challenges in accessing and negotiating the healthcare system [3]. Shoveller et al reported that factors involved in STI testing, such as urethral swab testing, were seen by men as fearful, and as a process where men felt very vulnerable; most of these male participants reported feeling anxious waiting for potentially bad news in addition to feeling that their masculinity had been compromised through an admission of needing help [4]. Limitations of these studies are that they did not specify whether patients had symptoms, or if men had experienced (invasive or non-invasive) testing procedures, and tended to recruit patients who had already been diagnosed with an STI, which makes the impact of testing and receiving a diagnosis difficult to separate [5,2–4]. Men who used postal screening for CT and had negative test results were not found to experience a negative impact on self-esteem or anxiety, suggesting that STI testing at home may reduce negative impacts associated with visiting an SHC, however this study did not follow up men with positive CT diagnoses [6]. Future studies should consider including asymptomatic men alongside symptomatic men, to better evaluate the impact of attending clinic, and the experience of having genital examination and invasive testing or not, on psychosocial wellbeing and self-esteem. "

R1Q3) Some men who were symptomatic (which by definition they had to be to meet the inclusion criteria) did not ultimately have any infection. It would be helpful at some point in the paper to briefly discuss alternative reasons for why these men had symptoms to clarify that there are other possible reasons for them.

We addressed this in the Results (page 6, lines 31-34)

"Men who reported urethral symptoms but had no abnormalities on urethral sample testing and negative CT and NG results (categorised as NAD) had a variety of non-STI causes for their symptoms

diagnosed, including irritant or Candida balanitis, lower urinary tract symptoms which were likely prostatic in origin, non-infectious genital pain syndromes, and symptoms secondary to urethral trauma."

R1Q4) T-tests to compare differences between groups both in terms of non-response bias, and the distribution of other characteristics were not significant, but this is not particularly surprising given the small sample size and subsequent low power- mentioning this as a limitation and discussing how power to detect differences between groups will increase if larger numbers are recruited in future studies would be beneficial.

We addressed this in the Discussion (page 9, lines 32-36)

"We did not find significant differences between groups in terms of non-response bias and other characteristics, however this is not surprising as the power to detect differences between groups is greatly affected by sample size and our study included a small number of participants. Recruiting larger numbers of participants in future studies will increase the statistical power to detect any important differences between groups."

R1Q5) Continuing from the point above on power- It would be my suspicion that participants with positive diagnoses would be less likely to continue with the study (as you state in your qualitative work)- as far as I can tell all diagnoses have been compared individually in a t-test to NAD. Combining all diagnoses into one group and comparing to NAD in one t-test may provide slightly more power to detect whether those with a positive diagnosis of any kind are less likely to continue with the study, which may highlight differences in response bias. The influence may of course be different depending on the diagnosis, but this could help in obtaining an overall picture.

The scope of this work was to compare each diagnosis, particularly specific diagnoses(CT and NG) compared to non-specific diagnoses (NSGI), hence combining them together would not have allowed us to understand the influence of a specific diagnosis in the psychosocial scores, and to compare baseline and follow-up groups. We have therefore not performed this additional analysis.

R1Q6) NAD patients also showed decreased self-esteem scores, which shows the possible importance of including asymptomatic men. It may be the clinic visit itself which is having an impact. A reference to past studies looking at the overall impact of a clinic visit on psychosocial/esteem outcomes with a brief discussion would be beneficial.

We addressed this query in response to R1Q2 – please see above.

R1Q7) Are you able to distinguish between those who answered the questions at each time point at the clinic, and who at home? Being at home, or otherwise using a device owned by the participant to answer the questions in a place of their choosing is thought to increase the likelihood of respondents reporting sensitive information. Looking at whether there is greater disclosure amongst those who did not answer in the clinic could provide valuable insight into how best to collect this information in the future. Given that this is a feasibility study which could influence the implementation of future research, this could be especially important.

We addressed this in the Results (page 7, lines 10-12)

"All patients who completed the baseline CASI did so using the Study tablet- device in a private room in the clinic. Participants completed the baseline CASI in about 5 minutes. All participants completed the follow-up CASI at home on a participant owned device".

R1Q8) It should be specified more often that it is heterosexual men to whom these results apply (e.g. in the abstract article summary, the first bullet point states this is a common diagnosis amongst men in general. If it is also common amongst men with other sexual identities further justification for selecting only heterosexual men should be given).

The justification for including only heterosexual men has been given in the response to R1Q1 above. The clarification that these results only apply to heterosexual men has been corrected in the text of the manuscript (please see comments marked R1Q8)

R1Q9) Table 1- I don't think the % are necessary with such small numbers. If you want to include them, it would be better to include in parentheses next to the n, so n(%). Otherwise the table is twice the width without adding any new information.

We have addressed this by putting the percentages in parentheses in Table 1. (Page 12)

R1Q10) Table 2- Which distribution has been used to calculate the p-values? For sample sizes under 30, students t-tests are conventionally used- have they been used here?

We addressed this in the Methods (page 6, lines 11-12)
"We used the students t-test distribution to calculate p-values".

We also added students t-test to the title of Table 2.a (page 13)

"Tab 2.a Distribution of psychosocial scores by final diagnosis within the baseline group and results from students t-test."

R1Q11) The STROBE form specifies inclusion of study design in the title or early on in the paper- I would suggest including the term 'observational study' somewhere in the abstract or early on in the paper to clarify the design.

We have now included the term observational study in the Abstract and Methods to clarify the study design.

See Abstract (page 2, lines 1-3)

"We conducted an observational study to evaluate the psychosocial impact of an NSGI diagnosis among symptomatic heterosexual men attending SHCs compared to the impact of a diagnosis of CT, NG or no abnormalities detected (NAD) focussing of the feasibility of our study methodology."

See Methods (page 4, lines 16-18)

"We conducted an observational, longitudinal study in symptomatic heterosexual men using patient completed questionnaires assessing HRQoL and psychosocial status before and after receiving STI test results, and individual qualitative interviews to evaluate experiences at the SHC and acceptability of study methods."

See Discussion (page 8, lines 27-29)

"In this small observational feasibility study we detected decreases in mean HRQoL and self-esteem scores in participants with a final diagnosis of CT or NSGI from the time of clinical presentation to approximately one week after receiving NAAT results."

Reviewer 2 (R2)

R2Q1) I wonder if the authors could explain a bit more about the rationale of the study. The authors mention in the introduction that there are no published data on the psychosocial impact of non-specific genital infection diagnosis among men. But what would be the relevance of having information about the psychosocial impact? In many medical conditions, a low (mental) health related quality of life is related with a poor retention in care and a poor adherence to medication recommendations including treatment adherence. Could that be of any relevance? Given that these men receive recommendations about taking antibiotics, inform their partners and to abstain from sex until treatment is completed.

We addressed this in the Introduction (page 3, lines 20-28)

"Men who undergo STI testing, and are diagnosed with an STI report negative impacts on their psychosocial wellbeing, including feelings of stigma [5,7,8], however, to our knowledge, there are no published data on the psychosocial impact of NSGI diagnosis among men. Stigma has been shown in men to be associated with a delay in testing for STIs, and decreased willingness to notify their casual partners of their STI diagnosis [9]. Studies on psychosocial impact, partner notification and quality of life impact have predominantly been carried out in women [6,8,10,11], with few studies recruiting men. A greater understanding of the degree and nature of psychosocial impacts of a diagnosis of an STI, or a 'presumed STI' such as NSGI, in men, is important to optimise interventions, including partner notification and risk-behaviour modification, which aim to reduce the negative impact of STIs, such as complications of infection.

R2Q2) The authors compared health-related quality of life and self-esteem between men who received a diagnosis of non-specific genital infection, Chlamydia trachomatis, Neisseria gonorrhoeae and no-abnormality detected. Did the authors have a-priori expectations about different impacts of receiving these diagnoses? Did they expect that some diagnosis would have more impact than others? Please explain."

We addressed this in the Methods (page 5, lines 30-33)

"We did not have prior expectations about different impacts of these diagnoses, there are no studies comparing the impact of CT to NG or NSGI in men, this was exploratory. We were interested in investigating whether NSGI was perceived as a benign diagnosis or whether it elicited negative psychosocial impacts which have been reported in men diagnosed with STIs such as CT [5]."

R2Q3) It was unclear to me how the scores for the EQ-5D-5L were analyzed because the authors presented a single mean score for this questionnaire. The EQ-5D-5L consist of 5 dimensions of quality of life that are scored on a 5-point Likert scale and a visual analogue scale. A utility score can be calculated on the basis of the scores on the 5 dimensions. Where does the mean EQ-5D score refer to? One of the dimensions, the VAS, the utility score or something else? Please explain.

We addressed this in the Methods (page 6, lines 2-6)

"To enable us to compare the EQ-5D-5L score across the different diagnoses (CT, NG, NSGI and NAD), a mean score was calculated: we calculated mean values for each of the five dimensions of the EQ-5D-5L, and a final score for the EQ-5D-5L was created using the sum of the mean value for each dimension. We calculated the mean value rather than an index score for the EQ-5D-5L to allow comparison between the outcomes of EQ-5D-5L with the Rosenberg Self-Esteem and Multidimensional Social Support Scales."

R2Q4) In the introduction section, the authors mention that one of the aims of this feasibility study was to inform their study methods. Despite the authors made many efforts to facilitate retention in the study, including giving financial incentives to participants, what would they do different (if anything) in

a next study to prevent attrition? What advice would they give to future researchers aiming to investigate a similar group of subjects?

We addressed this in the Discussion- see below

page 9, lines 2-18

"Despite concerns that the target group of heterosexual men attending SHCs would be hard to recruit [12], we found 75% of men were happy to enrol in this study. This should encourage researchers to investigate views from this group regarding STI diagnosis and screening. A study investigating the psychosocial experiences of patients attending a clinic, which recruited men and women diagnosed with an STI from an SHC, reported a recruitment rate of 33% [2]. This study involved ten interviews lasting between 20-45 minutes each. Our higher baseline recruitment rate may be due to the use of CASI interviewing, fewer interviews, or because patients were recruited before they had any diagnosis made.

One of the difficulties associated with conducting longitudinal studies is attrition. In our study, of those eligible for follow-up, 24 % overall completed the follow-up CASI, 25% of those with a diagnosis of NSGI, CT or NG, and 21% of those with NAD. A longitudinal questionnaire study investigating community screening for CT reported follow-up rates of over 60% , however, eligibility for follow-up excluded those with positive STI test results, and 60% of the sample was female, reflecting the higher response rate amongst women [6]. Our study's qualitative component had a follow-up rate of 27% overall, 25% in those who had any positive diagnosis, and 33% in those with NAD. By comparison, a study including men, investigating patient views on CT screening recruited 6.6% of those who completed the baseline questionnaire to a qualitative interview, which was below their target of 10% [13]."

page 9, lines 35-36

"Recruiting larger numbers of participants in future studies will increase the statistical power to detect any important differences between groups."

page 10, lines 28-30

"Future studies could also consider offering a qualitative interview at the point of recruitment, not only at follow-up, to widen the possibility of a qualitative exploration among participants who do not find the CASI questionnaire acceptable."

page 11, lines 17-32

"The issue of gender concordance between participants and interviewers/researchers has been discussed in the literature [14] and the fact that our study recruiter was female may have affected recruitment and retention. A study investigating attitudes to Viagra in heterosexual Hispanic men, using telephone qualitative interviews, found that changing from female to male recruiters and data collectors increased their recruitment [12]. Participants in our study were offered the option of a male or female interviewer but may not have felt comfortable with asking for a particular interviewer, and so this may not have been sufficient to reduce the potential impact of gender discordance on acceptance of the invitation to an interview in our study. We suggest future studies involving qualitative interviews should consider the gender of recruiters and interviewers.

We received feedback from our participants that some of the wording of questions on our CASIs did not appear to be directly relevant to our study aims. Future studies should ensure that data collection measures use terms that are clear and familiar to participants, possibly through development of questionnaires with focus groups. Some men in our study also saw participation as an opportunity to learn more about their own health and requested that future studies offer research participants

material on health promotion, i.e. material on the infection they were diagnosed with and how to reduce risk of infection."

References

- 1 Sonnenberg P, Clifton S, Beddows S, et al. Prevalence, risk factors, and uptake of interventions for sexually transmitted infections in Britain: findings from the National Surveys of Sexual Attitudes and Lifestyles (Natsal). *Lancet* (London, England) 2013;382:1795–806. doi:10.1016/S0140-6736(13)61947-9
- 2 Mulholland E, Van Wersch A. Stigma, sexually transmitted infections and attendance at the GUM Clinic: an exploratory study with implications for the theory of planned behaviour. *J Health Psychol* 2007;12:17–31. doi:10.1177/1359105306069098
- 3 Lindberg C, Lewis-Spruill C, Crownover R. Barriers to sexual and reproductive health care: urban male adolescents speak out. *Issues Compr Pediatr Nurs* 2006;29:73–88. doi:10.1080/01460860600677577
- 4 Shoveller JA, Knight R, Johnson J, et al. 'Not the swab!' Young men's experiences with STI testing. *Social Health Illn* 2010;32:57–73. doi:10.1111/j.1467-9566.2009.01222.x
- 5 Holgate HS, Longman C. Some peoples' psychological experiences of attending a sexual health clinic and having a sexually transmitted infection. *J R Soc Health* 1998;118:94–
6. <http://www.ncbi.nlm.nih.gov/pubmed/10076643> (accessed 10 Mar2016).
- 6 Mills N, Daker-White G, Graham A, et al. Population screening for Chlamydia trachomatis infection in the UK: a qualitative study of the experiences of those screened. *Fam Pract* 2006;23:550–7. doi:10.1093/fampra/cml031
- 7 Darroch J, Myers L, Cassell J. Sex differences in the experience of testing positive for genital chlamydia infection: a qualitative study with implications for public health and for a national screening programme. *Sex Transm Infect* 2003;79:372–3. <http://www.ncbi.nlm.nih.gov/pubmed/14573831> (accessed 10 Mar2016).
- 8 Kangas I, Andersen B, Olesen F, et al. Psychosocial impact of Chlamydia trachomatis testing in general practice. *Br J Gen Pract* 2006;56:587–93. <http://www.ncbi.nlm.nih.gov/pubmed/16882376> (accessed 10 Mar2016).
- 9 Morris JL, Lippman SA, Philip S, et al. Sexually transmitted infection related stigma and shame among African American male youth: implications for testing practices, partner notification, and treatment. *AIDS Patient Care STDS* 2014;28:499–506. doi:10.1089/apc.2013.0316
- 10 Gottlieb SL, Stoner BP, Zaidi AA, et al. A prospective study of the psychosocial impact of a positive Chlamydia trachomatis laboratory test. *Sex Transm Dis* 2011;38:1004–11. doi:10.1097/OLQ.0b013e31822b0bed
- 11 Kangas I, Andersen B, Olesen F, et al. Psychosocial impact of Chlamydia trachomatis testing in general practice. *Br J Gen Pract* 2006;56:587–93. <http://www.ncbi.nlm.nih.gov/pubmed/16882376> (accessed 27 Sep2016).
- 12 Jones SG, Pat Patsdaughter CA, Martinez Cardenas VM. Lessons from the viagra study: methodological challenges in recruitment of older and minority heterosexual men for research on sexual practices and risk behaviors. *J Assoc Nurses AIDS Care* 2011;22:320–9. doi:10.1016/j.jana.2010.10.009
- 13 Lorimer K, Reid ME, Hart GJ. "It has to speak to people's everyday life..." qualitative study of men and women's willingness to participate in a non-medical approach to Chlamydia trachomatis screening. *Sex Transm Infect* 2009;85:201–5. doi:10.1136/sti.2008.031138
- 14 Fenton KA, Johnson AM, McManus S, et al. Measuring sexual behaviour: methodological challenges in survey research. *Sex Transm Infect* 2001;77:84–
92. <http://www.ncbi.nlm.nih.gov/pubmed/11287683> (accessed 24 Mar2016).

VERSION 2 – REVIEW

REVIEWER	Pythia Nieuwkerk Academic Medical Center, Amsterdam, the Netherlands
REVIEW RETURNED	23-Oct-2017

GENERAL COMMENTS	The authors have adequately addressed most of my previous comments. However, I am still concerned about the statistical analysis of the EQ5D. Calculating the mean for each dimension and combining these means into an overall mean score is not supported by the manual of the EQ5D. https://euroqol.org/wp-content/uploads/2016/09/EQ-5D-5L_UserGuide_2015.pdf The manual emphasizes that It should be noted that the numerals 1-5 have no arithmetic properties and should not be used as a cardinal score (page 6). I suggest the authors analyze the EQ5D in a way that is supported by the manual.
---

REVIEWER	Sarah Burkill Karolinska Institutet Centre for Pharmacoepidemiology Sweden
REVIEW RETURNED	30-Oct-2017

GENERAL COMMENTS	All my concerns have been addressed
-------------------------------------

VERSION 2 – AUTHOR RESPONSE

1. E5-5D-5L Index value revision: changes are highlighted in yellow in the manuscript

Reviewer Name: Pythia Nieuwkerk

Institution and Country: Academic Medical Center, Amsterdam, the Netherlands

Competing Interests: None declared

The authors have adequately addressed most of my previous comments.

However, I am still concerned about the statistical analysis of the EQ5D.

Calculating the mean for each dimension and combining these means into an overall mean score is not supported by the manual of the EQ5D.

https://euroqol.org/wp-content/uploads/2016/09/EQ-5D-5L_UserGuide_2015.pdf

The manual emphasizes that It should be noted that the numerals 1-5 have no arithmetic properties and should not be used as a cardinal score (page 6).

I suggest the authors analyze the EQ5D in a way that is supported by the manual.

We have addressed this by calculating an EQ-5D-5L Index value as recommended by the manual https://euroqol.org/wp-content/uploads/2016/09/EQ-5D-5L_UserGuide_2015.pdf, for each study participant. We calculated a mean EQ-5D-5L Index value for each diagnostic category in order to compare changes in the EQ-5D-5L Index value between categories. We have also added further information in the manuscript to explain our rationale for choosing the EQ-5D-5L as a measure of health related quality of life.

Abstract, Results, Page 2, Lines 69-71

"Compared to baseline, among patients completing FU-1, only patients with a diagnosis of NSGI ($p<0.05$) or CT ($p<0.05$) showed increased EQ-5D-5L Index"

Abstract, Conclusions, Page 2, Lines 77-79

"We found that heterosexual men attending SHCs with symptoms of urethritis experience both positive and negative psychosocial impacts following their clinic attendance, which warrants further investigation."

Methods, Page 5, Lines 170-174

"The EQ-5D-5L is the preferred utility measure recommended by the UK National Institute for Health and Clinical Excellence [22] and has five dimensions: mobility, self-care, usual activities, pain/discomfort and anxiety/depression. Weighted UK preference values are linked to the self-reported health state scores for a 0–1 Index value, where 0 is death and 1 is perfect health."

Secondary outcomes, Pages 5, Lines 200-201

"Percentage change from baseline to FU-1 in mean EQ-5D-5L Index value, and mean RSES value in men diagnosed with CT, NG, NSGI or NAD."

Data analysis, Questionnaire data, Page 6, Lines 210-215

"For all eligible men, RSES score, MSPSS score, and EQ-5D-5L Index value were calculated according to published methods [24][23][25][27].

For men in each diagnostic category, paired t-tests were used to compare mean values for the EQ-5D-5L Index, RSES and MSPSS at baseline, and to compare differences in mean values for EQ-5D-5L Index and RSES score, from baseline to follow-up."

Results, CASI analysis, Lines 272-277

"EQ-5D-5L Index value at baseline did not vary significantly between diagnosis groups. We found men with NSGI and CT showed a significant increase in mean EQ-5D-5L Index value (indicating better HRQoL) between baseline and FU-1 (+17% ($p=0.006$) and +219% ($p=0.005$), respectively) (Table 2b). There was no change in mean EQ-5D-5L Index value for men diagnosed with NAD."

Discussion, Page 8-9, Lines 306-322

"In this small observational feasibility study we assessed patient reported self-esteem and HRQoL using validated scales in heterosexual men presenting to SHCs with urethral symptoms, before undergoing STI testing, and after receiving all test results. This is the first study to investigate the HRQoL and psychosocial impact of a curable STI diagnosis in symptomatic men attending SHCs, using validated scales, and is one of the few available with data on STIs and EQ-5D-5L [28].

We found increased EQ-5D-5L Index value (indicating increased HRQoL) in men diagnosed with NSGI or CT from baseline to FU-1. Published studies have reported that some men diagnosed with CT reported a lack of concern regarding the diagnosis as they perceive CT as a relatively minor

infection [8,10,14,32]. This was supported by one of our qualitative interview participants. The increases we found in the EQ-5D-5L Index value may reflect decreases in the anxiety and pain/discomfort domains of the scale from just prior to consultation at baseline to receipt of all test results in FU-1. Shoveller et al. reported that the majority of men diagnosed with an STI in SHCs had reported feeling anxious waiting for potentially bad news [31], and so this finding may reflect that men found relief following treatment and explanation of their symptoms, and receiving negative results for more serious infections such as HIV. Due to our small sample size we were unable to find significant associations between participant satisfaction with their clinic visit and relief from symptoms and infection status in our CASI FU-1 data."

Discussion, Page 9, Lines 334-340

"Limitations

Our choice of HRQoL tool, the EQ-5D-5L, may not be sensitive enough to detect impacts caused by the diagnosis of CT, NG or NSGI, which are unlikely to cause problems in at least two of the five domains: mobility, and self-care. Studies investigating HRQoL and sexual health have found few significant differences with comparator groups using generic QoL instruments alone [33]. STI-specific HRQoL and psychosocial impact tools have been developed for genital herpes and genital warts [34–36] and several studies have now also combined both EQ-5D-5L and STI-specific tools [37–40]."

Discussion, Page 12, Lines 436-442

"Conclusions

Our study found that heterosexual men with urethral symptoms may experience a range of psychosocial and HRQoL impacts following SHC attendance and STI testing. More work is needed to investigate the most appropriate methodology for investigating sexual health and QoL, particularly around choosing between condition specific and generic measures, and reducing loss to follow-up in longitudinal studies. Future research in this area is needed to test the generalizability of our findings as to whether there are significant psychosocial harms of giving presumptive and non-specific diagnoses."

Table 2. Page 15

2. Rosenberg Self-Esteem Scale (RSES) revision: changes are highlighted in green in the manuscript Abstract, Results, Page 2, Lines 71-72

"patients with a diagnosis of NSGI (P=0.05) showed decreased mean Rosenberg Self-Esteem Scale score."

Methods, Questionnaire design, Page 5, Lines 174-176

"The RSES is widely used in social science research to measure self esteem; it uses a scale of 0-30 where a score less than 15 may indicate low self esteem, 15-25 normal self-esteem and 25-30 high self-esteem."

Results, CASI analysis, Page 7, Lines 271-272

"Participants with NG also had lower mean RSES scores (indicating lower self-esteem) at baseline as compared to those with NSGI (p=0.035)."

Page 8, Lines 277-278

"In contrast, we found decreases in mean RSES scores from baseline to FU-1 in men with NSGI (-30%, p=0.05)."

Discussion, Page 9, lines 324-332

"Furthermore, in men diagnosed with NSGI, we found decreased mean RSES score (indicating decreased self-esteem) from baseline to approximately one week after receiving all STI test results."

These findings support data from previous qualitative studies where men diagnosed with an STI in an SHC reported negative psychosocial impacts including stigma; anxiety; shame; isolation; concerns regarding relationships; a loss of social status; vulnerability; a lack of privacy and fear of STI testing, particularly urethral swab testing [8,10,29–31]. We did not find a decrease in self-esteem in men diagnosed with CT which might suggest that NSGI is perceived differently to CT. In our qualitative interviews participants with NSGI reported concerns regarding the uncertainty of the diagnosis, and fear of the impact of a possible STI diagnosis on their relationships; these factors may be important in the observed decreased self-esteem in these men."

Table 2, Page 15

Table 2 Distribution of psychosocial scores by final diagnosis

3. Multidimensional Scale of Perceived Social Support (MSPSS) revision: changes are highlighted in blue in the manuscript

Methods, Questionnaire design, Page 5, Lines 179-181

"The MSPSS is widely used to assess perceived social support and consists of three domains: family, friends, and significant others. Mean MSPSS scores from 1- 2.9 may be considered low support; 3- 5 moderate support and 5.1- 7 high support [26]."